# Do behavioural risks cluster among college students in Chandigarh, India? Novel insights from a latent class analysis

Vikas Kumar Bhatia[1][☯], Adhish Kumar Sethi[1][☯], Pratistha Sharma[1], Shubh Mohan Singh[2], Pinnaka Venkata Maha Lakshmi[1]*

1 Department of Community Medicine and School of Public Health, Postgraduate Institute of Medical Education and Research, Chandigarh, India, 2 Department of Psychiatry, Postgraduate Institute of Medical Education and Research, Chandigarh, India

☯ Vikas Kumar Bhatia and Adhish Kumar Sethi contributed equally, as joint first authors.
* pvm_lakshmi@yahoo.com

## Abstract

### Background

Youth is a critical phase in life, as behavioural risks in youth can have profound health impacts throughout the life course. Coexistence of behavioural risks is especially hazardous, and is important to address in public health interventions. Given the gaps in existing research on this issue, we aimed to determine the prevalence of multiple behavioural risks and identify their clustering among young adults in a prominent North Indian city.

### Methods

We collected data from young adults aged 18–22 years in a representative survey conducted across six colleges, assessing injury risks, victimisation, suicide, substance use, diet and physical activity using a self-administered questionnaire. We determined the prevalences of individual risks. We then identified risk clustering using latent class analysis, with gender as a covariate. We selected the most appropriate latent class model based on fitted probabilities, likelihood ratio tests, entropy, Akaike information criterion and Bayesian information criterion.

### Results

Of 752 participating students with median age 19 years, 64.4% identified as female. Latent class analysis identified four classes: multiple risks (8.8%), smoking and alcohol-related risks (5.4%), only dietary and physical activity risks (73.7%), and victimisation and injury risks (12.1%). Unhealthy diet and insufficient physical activity were the most common risks, and occurred uniformly across classes. Men were more likely than women to belong to higher-risk classes.

**Data availability statement:** All relevant data are within the manuscript and its Supporting Information files.

**Funding:** The author(s) received no specific funding for this work.

**Competing interests:** The authors have declared that no competing interests exist.

## Conclusion

Policymakers working for youth health must consider the interconnectedness of behavioural risks and their gender differentials, to simultaneously address multiple risk factors such as violence, unsafe sex and substance use.

## Introduction

The most valuable human resource for promoting a country's political, cultural, and economic development is seen to be its youth, who are the most productive and dynamic segment of the population [1]. The United Nations defines 'youth' as the age group of 15–24 years [2]. This age is the phase of change and development of new behaviours [3]. People at this age are particularly vulnerable to developing behavioural risks. While definitions vary, 'behavioural risks' are commonly conceptualised as avoidable actions, or omissions, of individuals which increase chances of adverse health outcomes for themselves or others [4]. Some examples of behavioural risks which often begin in youth are unsafe driving, tobacco use, alcohol use, drugs, violence, unsafe sexual behaviours and unhealthy diets [5].

About a fifth of the world's youth live in India [6]. Behavioural risks are an important contributor to morbidity and mortality among youth in India. This is elucidated by findings from the Global Burden of Disease study, published in 2021. In 2019, unintentional injuries and transport accidents together accounted for a quarter of the deaths in the 10–24 years age group in India, with self-harm and violence accounting for another 17% of the deaths in this age group [7]. In that year, alcohol use was found to be the most important behavioural risk for death in the 10–24 years age group, accounting for 2.6% of total deaths. Similarly, unsafe sex accounted for 0.9% of deaths and tobacco use for 0.3% of deaths [7]. Many of the deaths in transport accidents in India are attributable to behaviours such as non-use of seat belts and helmets [8]. Unhealthy diets and insufficient physical activity, when established during youth and persistent in the long term, show their effects late in the life course, in the form of non-communicable diseases like cardiovascular disease [9].

Considerable research has been done on the prevalence of different behavioural risks among youth in India [10–13]. According to the fifth round of the National Family Health Survey (NFHS–5), 14.3% men and 0.8% of women aged 15–19 years used tobacco. In the same age group, 5.8% of men and 0.2% of women used alcohol, 27.1% of men and 16.9% of women took aerated drinks at least once a week, and 0.6% of men and 0.1% of women had sexual intercourse with two or more partners in the last 12 months [14]. The survey also found marked regional variations in the prevalences of these behavioural risks, which can be explained by the diverse socio-cultural conditions in different states of India [14]. The second round of the Global Adult Tobacco Survey (GATS–2) 2016–17 found that 12.4% of persons aged 15–24 years in India were current tobacco users. Again, regional heterogeneity was noted, with a comparatively low prevalence in the northern states of India [15]. Similarly, a study in Bhubaneshwar

found that almost half of college students did not follow a regular exercise routine [16]. Sexual activity in youth is rare in India, as compared to other countries [17]. Yet, among those who are sexually active, the prevalence of condomless sex may be quite high. This is particularly true for boys and young men. A study conducted in Chandigarh in 2020 found that nearly two-thirds of sexually active college students had unprotected sex [18]. Another study among men aged 18–24 in Ballabgarh, Haryana, found that almost one-third of those sexually active had engaged in condomless sex [10]. Use of psychoactive substances before sex is another concern, as it impairs judgement and decreases condom use [19]. There are limited data on this practice among youth in India [20]. The literature also provides evidence on the prevalence of suicidal ideation [21–23] and bullying [24–26] among youth in India, though these studies do not provide adequate information about the scenario in northern India.

Also, most studies on behavioural risk factors among youth in India have concentrated on specific risk factors in isolation. More attention is now being given to the co-existence of multiple risks in the same individual [27,28]. Such co-existence arises from shared biological, social and environmental determinants of different behaviours. Identifying such syndemics of behavioural risk is important for two reasons. First, where an individual is exposed to multiple risk factors, interaction of risk factors can greatly elevate health risks [29–31]. Second, identification of groups with clustering of risk factors can help in efficiently targeting multiple risks simultaneously, through composite intervention packages. [32] However, we found limited data on clustering of multiple behavioural risks among youth in India [27].

Therefore, we planned this study to determine the prevalence of multiple behavioural risks among youth in a north Indian city, and identify clustering of risks within individuals, if any. We focused on youth attending colleges for higher education. We did this for several reasons. One, colleges were sites where we could readily access eligible individuals, using the limited resources that we had for this study. Two, colleges are an important avenue for youth health promotion programmes [33,34]. Three, in the Indian context, higher education is a critical phase where the increased autonomy from parental control means that a person is likely to adopt unsafe behaviours [35,36]. College students may show greater prevalence of behavioural risks, such as binge drinking, than their non-college peers [37].

Though some studies have been done on behavioural risks among college students in Chandigarh [38,39], we could not find any study that assessed co-occurrence of a comprehensive list of behavioural risks. Hence, we considered it a suitable site for the study.

## Methods

### Study setting and design

This was a cross-sectional study conducted in colleges in Chandigarh. We recruited participants from July 2 to August 30, 2018. The city of Chandigarh is located in the northern part of India, and is notable for being a higher education hub, with a university and 26 colleges including over 80 departments [40]. Its student body comprises individuals hailing from northern India as well as other parts of the country [41].

### Study population and eligibility criteria

The study population consisted of students aged 18–22 years enrolled in colleges in Chandigarh who provided consent for the study. There were no exclusion criteria.

### Study size and sampling strategy

We calculated the study size required to estimate the prevalence of current smokeless tobacco use in the study population, with a relative precision of 20% and a confidence level of 95%. Assuming the population prevalence to be 10.8% as per the GATS–2 results for the 15–24 years age group in India [15], we needed 793 respondents. Assuming a non-response rate of 20%, we needed to invite 992 individuals for participation.

There is no straightforward way of calculating study size for latent class analyses [42]. However, using a rule of thumb of 300–1000 individuals for model fit indices to perform adequately, we considered that our study size would be adequate for latent class analysis [43].

We used multistage random sampling to get a representative sample of college students in Chandigarh. Out of 32 government and private colleges in Chandigarh, six colleges were randomly selected—one women's college and five co-educational colleges. This ratio was taken keeping in mind the relative numbers of students in these colleges. If a selected college refused to participate, then it was replaced by another college chosen randomly from the same stratum. The colleges had three educational streams: Science, Arts and Commerce. For sampling within colleges, we considered the educational streams as strata. Within streams, students were divided into sections, which attended classes together. From each stream, we selected 3–4 sections randomly. All students in the selected sections, i.e., about 150 per college, were invited for participation.

## Study tool

The data were collected using a self-administered paper-based structured questionnaire in the English language. This was an adapted version of the 2017 Youth Risk Behavior Surveillance System (YRBSS) Standard High School question-naire from the Centers for Disease Control and Prevention (CDC) [44], with a few changes to reduce respondent burden and make it relevant to the Indian scenario. The specific changes were as follows. We enquired about 'college year' instead of 'grade'. We removed the items on race/ethnicity and gun violence, as we did not consider these sufficiently relevant in the local cultural context. In the section on 'other tobacco products', we retained only the items on smokeless tobacco products, as other tobacco types are extremely rare in this context. For smokeless tobacco products, we used terms in the local language. Similarly, for cannabis and cocaine, we used terms from the local language to improve com-prehension. We removed the items on concussion, HIV, dental care and asthma to reduce respondent burden.

The questionnaire was pilot tested among 10 Masters' students in the Institute where the investigators were working. This Institute was not in the list of colleges from which sampling was done for the main study.

The items in the questionnaire encompassed the demographic details of participants, self-reported height and weight, and behavioural characteristics such as risk factors for injury, depressive symptoms, suicidal thoughts and attempts, substance use, dietary habits, physical activity, sleep and sexual behaviour. In total, the questionnaire comprised 74 questions. To maintain anonymity, it did not ask for the participant's name or address. Most of the questions involved selecting responses from given options. Only the height and weight questions involved writing numerical digits in the data collection form. Hence, it was not possible to identify students by their handwriting. The complete questionnaire is provided in S1 File.

## Data collection

After obtaining necessary permissions from the college principal, one of the investigators visited each college on a work-ing day. He visited the classrooms where the selected sections of students studied, provided information about the nature and purposes of the study, and invited students to participate. He then distributed the questionnaire and a written consent form to all students in the classroom, and gave standardised instructions on filling the questionnaire. The investigator was present throughout the data collection time to explain any question which a participant was unable to understand. After about 45 minutes, the investigator collected the filled questionnaires.

## Statistical analysis and reporting of results

We entered the data into a spreadsheet, and then imported the spreadsheet to statistical analysis software. We did data processing, cleaning and prevalence estimation with R 4.3.0 (R Foundation for Statistical Computing, Vienna, Austria).

First, we examined the data to look for any implausible values or logically inconsistent responses (e.g., a partici-pant who reported never using cigarettes, but specified the age at which they started smoking cigarettes). We set such

implausible or logically inconsistent values to missing, in line with the CDC YRBSS methodology [45]. However, for each such individual, we retained data for all other responses. Individuals with valid data for <20 questions were excluded from the analysis.

Then, we summarised the socio-demographic characteristics of the participants using numbers and proportions. Age was summarised using median and interquartile range (IQR). We thenlisted 27 behavioural risks for which we had data available, and which we considered sufficiently important, based on the literature on health risks among youth in India [7]. While shortlisting these 27 risks, we tried to ensure diversity in terms of the items covered, while maintaining parsimony of items for latent class modelling. Accordingly, we omitted items which we found extremely rare (e.g., solvent and heroin use) or of uncertain importance for public health (e.g., dairy consumption). Also, where multiple items measured nearly identical constructs, we included only one. For example, 'faced sexual abuse during dating in the last 12 months' was highly correlated with 'faced sexual abuse in the last 12 months,' so we kept the latter only. We recoded these 27 behavioural risks as binary variables. For ease of interpretation, we also organised them into seven broad domains: injury risks (four items), victimisation (four items), depression and suicide risk (two items), substance use (five items), nutrition and diet (four items), physical activity, sedentary behaviour and sleep (five items), and sexual behaviour (three items).

Body mass index (BMI) was calculated based on self-reported height and weight, as weight in kg/(height in m)$^2$. We considered a BMI of 25 kg/m$^2$ or more as overweight/obese, in line with WHO cutoffs [46]. We defined 'current use' of a substance (cigarettes, smokeless tobacco, alcohol or cannabis) as use of that substance in the last one month. We defined 'binge drinking' as consuming four or more drinks at a time for women, and five or more drinks at a time for men. We defined 'insufficient physical activity' as being physically active for at least 60 minutes a day for less than five days in the last week. For each behavioural risk, we calculated prevalence as a proportion with its 95% confidence interval (CI).

To identify clustering of risk factors, we performed latent class analysis using MPlus Version 8.4 (Muthen & Muthen). We included the 27 behavioural risks as binary variables, with gender as a covariate. We ran models with two to seven latent classes. We examined the fitted probabilities, entropy, p-values from the Vuong–Lo–Mendell–Rubin likelihood ratio test, the Akaike information criterion (AIC), Bayesian information criterion (BIC) and sample size-adjusted Bayesian information criterion (aBIC) for each model, to select the model with best performance. For this selected model, we reported probability of belonging to each latent class, probability of having each risk factor within each latent class, and the multinomial odds ratios for association of gender with latent class membership. We also assigned short, descriptive labels to each class, based on the model estimates. We labelled classes according to the behavioural risk domains which showed greatest within-class fitted prevalence. Though we had included rare risk factors (e.g., suicide attempts, sexual risks) in the models, we avoided including them in class labels, as their parameter estimates could be statistically unstable.

### Ethical considerations

The Institute Ethics Committee of the Postgraduate Institute of Medical Education and Research, Chandigarh approved the study protocol (letter number INT/IEC/2018/000791, dated May 24, 2018). Participants provided written, informed consent on paper-based consent forms, and investigators made all possible attempts to maintain privacy and confidentiality of the data. Though the consent forms bore the signatures and names of participants, it was not possible to link them with the data collection forms which the participants submitted, thus ensuring anonymity. Given the anonymous nature of data collection, it was not possible for us to link participants with care if they reported behavioural risks (e.g., suicidal ideation or substance use).

## Results

### Participant recruitment

Of the 995 eligible students invited, 190 students did not consent for participation and 53 returned questionnaires with fewer than 20 questions answered. In all, 752 participants returned completed questionnaires, giving a response rate

of 75.6%. After removing implausible and logically inconsistent values, we had complete data for 606 individuals. The remaining 146 participants had missing data for a median of six questions each. They were included in the analysis only for the questions to which they provided valid responses.

## Socio-demographic profile

Among the 752 respondents, 268 (35.6%) identified as male, 484 (64.4%) identified as female, and none as transgender. The median age of the respondents was 19 years (IQR 19–21 years).

## Prevalence of behavioural risks

The prevalences of different behavioural risks, along with their 95% confidence intervals, are shown in Fig 1 and S2 File. The greatest prevalences were noted for nutritional and dietary risks: 78.2% (95% CI [75.0%, 81.1%]) did not eat vegetables at least twice a day in the last week, 70.1% (95% CI [66.6%, 73.3%]) did not eat fruits at least once a day in the last week, 15.7% (95% CI [13.2%, 18.5%]) had aerated drinks every day in the last week, and 14.5% (95% CI [12.1%, 17.3%]) were overweight/obese.

Risks pertaining to physical activity and sleep were also found to be quite common, with 59.8% (95% CI [56.2%, 63.4%]) reporting insufficient physical activity, 59.2% (95% CI [55.6%, 62.7%]) reporting muscle strengthening exercises on less than two days in the last week, and 46.0% (95% CI [42.4%, 49.7%]) reporting average nighttime sleep of less than seven hours.

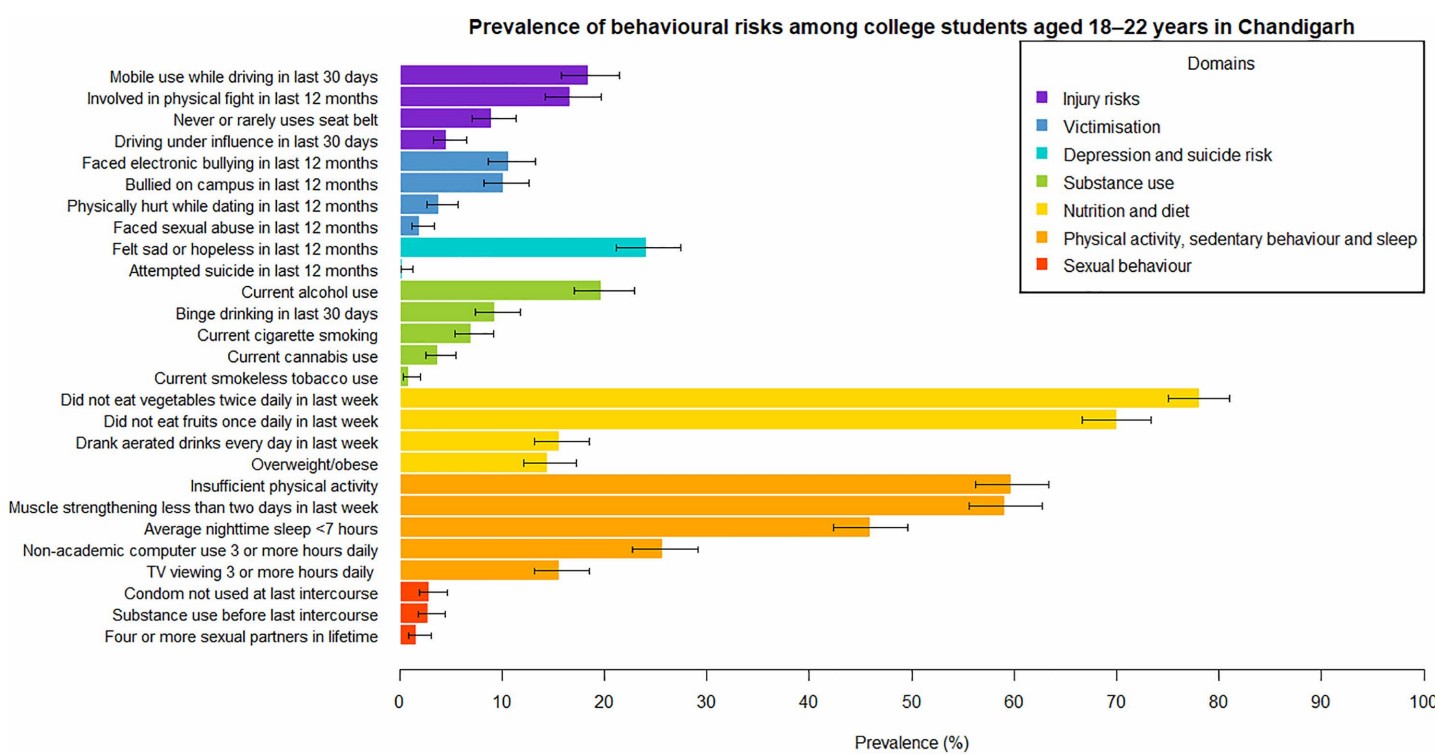

**Fig 1. Prevalence of behavioural risks among college students aged 18–22 years in Chandigarh.** The bars show the point estimates, and the error bars show the 95% confidence intervals. The denominators for the prevalences ranged from 701 to 752, depending on the number of individuals for whom data were available.

The prevalences of current substance use were 19.8% (95% CI [17.0%, 23.0%]) for alcohol, 7.1% (95% CI [5.4%, 9.2%]) for cigarettes, 3.8% (95% CI [2.6%, 5.5%]) for cannabis, and 0.9% (95% CI [0.4%, 2.0%]) for smokeless tobacco.

Three percent (95% CI [1.9%, 4.6%]) of participants reported being sexually active with non-use of condoms at their last sexual intercourse.

### Latent class analysis

With our specified items, and gender as covariate, we were able to fit satisfactory latent class models with two to five classes. We selected the model with four classes as it was able to distinguish meaningful classes, had a relatively high entropy (0.916), and reasonably low AIC, BIC and aBIC (Fig 2 and S3 File). Also, the Vuong–Lo–Mendell–Rubin likelihood ratio test showed substantial improvement in fit up to four classes, but not on adding a fifth class (p = 0.366 for five versus four classes). Fig 3 and S4 File show the pattern of item probabilities within these four classes. Some of the risks, namely

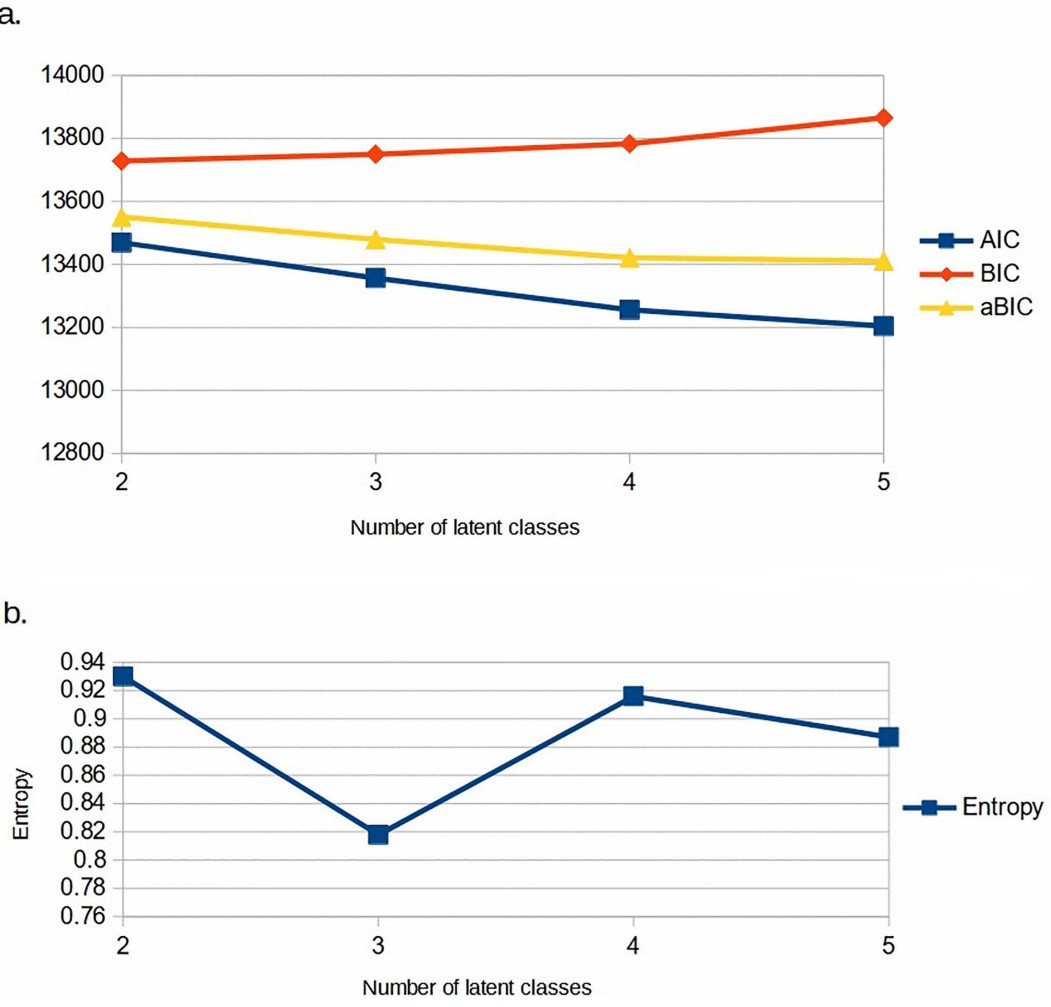

**Fig 2. Assessment of latent class models for behavioural risks among youth in Chandigarh.** Model fit was assessed using **a.** AIC, BIC and aBIC, and b. entropy, for models with different numbers of classes.

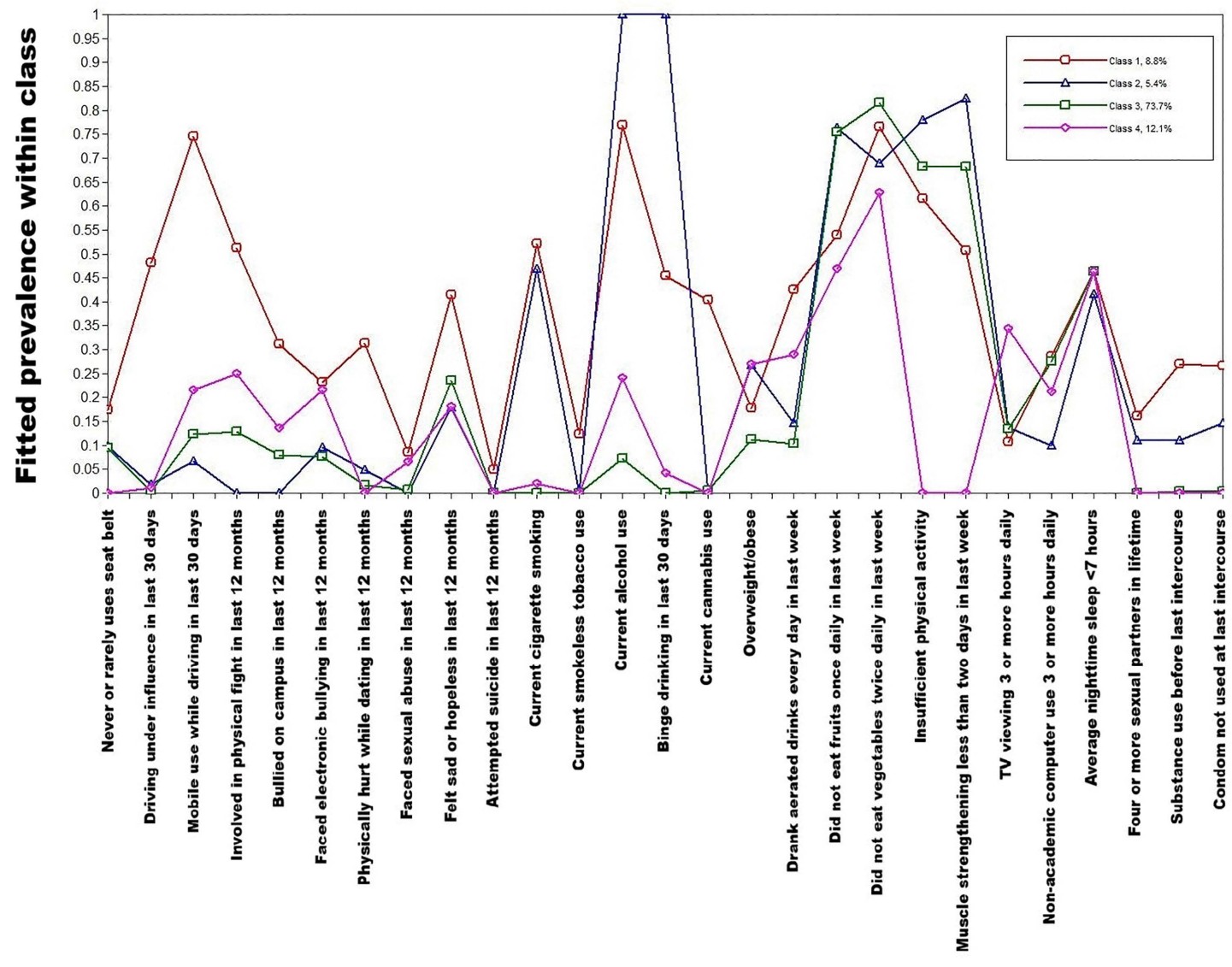

**Fig 3. Results of the model with four latent classes, for behavioural risks among college students aged 18–22 years in Chandigarh.**

nutritional risks, digital device use and sleep, did not differ appreciably across the classes. The other risks showed a distinctive pattern across classes.

Class 1 (estimated to constitute 8.8% of the population) was characterised by high endorsement of risks in almost all domains. These included driving under influence of alcohol (48.2%), mobile use while driving (74.4%), involvement in physical fights (51.1%), depressive symptoms (41.5%), cigarette smoking (52.1%), smokeless tobacco use (12.4%), alcohol use (76.8%), cannabis use (40.3%), substance use before last sexual intercourse (26.9%), and condomless sex (26.6%). Hence, we labelled this class as 'multiple risks'.

Class 2 (5.4%) was marked by high endorsement of current cigarette smoking (46.8%), current alcohol use (100.0%), binge drinking (100.0%), four or more sexual partners in lifetime (11.0%), substance use before last sexual intercourse

(10.9%), and condomless sex (14.6%). This class had low estimated prevalence of driving-related risks, physical violence and bully-victimisation. Accordingly, we labelled it as 'smoking and alcohol-related risks'.

In contrast, individuals in Class 3 (73.7%) had low estimated risk in most domains. Their prominent risks were diet, physical activity, digital device use, and sleep. Hence, we labelled Class 3 as 'only dietary and physical activity risks'.

Finally, Class 4 (12.1%) was distinguished by high estimated prevalence of mobile use while driving (21.6%), involvement in physical fights (25.0%), bully-victimisation on campus (13.5%), and cyber bully-victimisation (21.6%). This was the only class with low prevalence of insufficient physical activity (0.0%) and inadequate muscle strengthening exercises (0.0%). We labelled this class as 'victimisation and injury risks'.

The multinomial odds ratios for class membership, comparing women and men, are shown in Table 1. Women were less likely than men to belong to all three higher-risk classes.

## Discussion

### Summary of results

In this cross-sectional study among college-going youth in Chandigarh, we observed a wide variety of behavioural risks that could adversely affect the health of youth over the life course. The most notable among these were risks related to nutrition and physical activity, with over two-thirds of youth reporting low consumption of fruits and vegetables, and over half reporting insufficient physical activity. Interestingly, these risks occurred almost uniformly across the latent classes. At the same time, we also found distinct classes for other behavioural risks among youth, with a substantial proportion being at low risk and smaller proportions showing clustering of specific risk factors, such as violence, substance use and unsafe sexual behaviour. Our findings also underscore significant gender disparities in behavioural risks, with men demonstrating higher odds for behavioural risks. This disparity was particularly marked for Class 1, i.e., the latent class with a combination of injury, substance use and sexual risks.

### Physical activity and nutrition

The World Health Organization recommends 60 minutes of moderate-to-vigorous physical activity daily for adolescents, and 150–300 minutes per week for adults [47]. Data from multiple parts of the country consistently show that one-third to two-thirds of adolescents and young adults fail to meet the guideline values for physical activity [16,39,48,49]. In our study, almost 15% of participants were overweight or obese, based on self-reports of height and weight. A study among university students in Delhi during 2018–20 found similar results [50]. NFHS–5, a large, nationally representative survey conducted in India from 2019 to 2021, also shows that overweight and obesity in this age group are important concerns throughout India, with large cities such as Chandigarh and Delhi bearing a greater burden [14]. This is likely due to the high socio-economic status and obesity-promoting environment in these locations. Our study also explains some of the causes of physical inactivity and overweight/obesity among youth. We observed that one-fourth of participants used computers for non-academic purposes for three or more hours per day, and one-sixth watched television for three or more hours a day. Such sedentary behaviour takes away time from physical activity, and predisposes to obesity [51]. We also

**Table 1. Multinomial odds ratios for membership in latent classes of behavioural risks, for women as compared to men.**

| Class | Odds ratio | 95% CI for odds ratio | p-value |
|---|---|---|---|
| Class 1 (multiple risks) | 0.08 | (0.04, 0.16) | <0.001 |
| Class 2 (smoking and alcohol-related risks) | 0.09 | (0.04, 0.22) | <0.001 |
| Class 4 (victimization and injury risks) | 0.17 | (0.09, 0.34) | <0.001 |

Note: Class 3 (only dietary and physical activity risks) was the reference category for each of the other classes.

observed a high intake of "empty calories", in the form of aerated drinks, among our study participants. There is a need for concerted action on these proximal determinants of non-communicable diseases, by creating an enabling environment for physical exercise [48], and taxation and sales restrictions of unhealthy food and beverages [52].

### Sexual behaviour

Our study found small but important prevalences of unsafe sexual practices, including multiple sexual partners, substance use before sex and condomless sex, among youth in Chandigarh. Our observed prevalence of condomless sex was similar to that in NFHS–5, but lower than that observed in similar age groups in other studies in India [10,14,17,18]. This could be due to higher educational attainment and greater awareness of sexually transmitted infections among our study participants. Our result could also be biased downward by social desirability, as participants in our study filled the questionnaires in classrooms, and might have been concerned about sensitive information being disclosed to peers despite measures to maintain privacy. Our study yields novel data on substance use before sex, an emerging risk factor for HIV and sexually transmitted infections among youth in India [20,53] India's National AIDS and STD Control Programme includes Red Ribbon Clubs for promotion of HIV preventive behaviour in colleges [54]. Our study suggests that the activities of these clubs need to be intensified, particularly for prevention of substance use preceding sex, which is a relatively neglected area.

### Suicide

Suicide is an increasingly important cause of death among youth in India [55]. In our study, 0.4% of students reported suicide attempts in the past 12 months. This is lower than that in most Indian studies, but higher than the prevalence in other parts of the world [10,21–23,56]. Our findings suggest that, though not as common as in other parts of India, suicidal ideation and behaviour need specific attention in mental health programmes in Chandigarh. The National Mental Health Programme and National Suicide Prevention Strategy in India recognise the importance of colleges as a platform for suicide prevention interventions [57,58]. Their implementation at college level needs to be strengthened to achieve the desired impact.

### Latent class analysis

Our analysis highlights that behavioural risks were not randomly distributed among youth in Chandigarh. Rather, over two-thirds of the individuals in our study had few risk factors, while about 9% faced multiple risks. This clustering of risks can be explained by an interplay of biological, psychological, and environmental factors. Unsafe sexual behaviour and psychoactive substance use stimulate common brain areas and produce analogous feelings of reward or thrill [59]. Thus, the coexistence of these behaviours can be explained by a temperament for reward-seeking among certain individuals [60]. The co-occurrence of risky driving and physical fights with bully victimisation can be considered psychologically as a defence mechanism, wherein the individual "acts out" to relieve the stress caused by victimisation [61]. Similarly, the co-occurrence of victimisation, depression, and suicidal ideation, as seen in Class 1 in our study, has been noted in earlier work [62]. Indeed, these constructs might be causally linked, as shown by previous studies. For example, longitudinal studies show that cyber-bullying victimisation among adolescents correlates with subsequent depressive symptoms [63,64].

We observed a greater tendency of men to have behavioural risks. While we did not explore mechanisms for this difference, previous work shows that biological differences between men and women could play a role. The male hormone testosterone is known to promote aggression and risk-taking [65]. This might be reinforced by social norms which validate risky endeavours by men and discourage such endeavours by women [66]. Uncovering the relative contributions of nature and nurture to gendered risk-taking could be an idea for future research in India.

Another study conducted among adolescents aged 14–19 years in New Delhi found similar results, i.e., low physical activity tended to coexist with low fruit and vegetable intake, while tobacco use tended to co-occur with alcohol use. That study also noted that males and individuals with lower socio-economic status were more likely to have multiple risk

factors. However, that study used a different statistical technique, i.e., cluster analysis, to identify clusters, while we used latent class analysis [27].

In contrast, other studies done among youth in India have used cluster analysis, and covered nutritional risks. An analysis of a nationally representative survey conducted among adolescents aged 10–19 years in India in 2016–2018 identified five clusters based on dietary patterns, metabolic risks and micronutrient deficiency. These clusters were "comparatively healthy", "plant-based", "obesogenic diet", "Western diet" and "convenient" [67]. Likewise, a study conducted among adolescents aged 10–19 years in Bihar and Assam identified plant-based and mixed diets as two distinct clusters, with males being more likely to take a plant-based diet [68]. Our study was novel in that we used latent class analysis, a flexible approach which does not require specification of any "distance" or "dissimilarity" measure to identify similar individuals. Our assessment of lifestyle risks was also quite wide in scope as compared to previous studies, as we included injury risks, suicide, substance use and sexual behaviour.

There is varied literature on clustering of behavioural risks among youth in other countries of South and South-East Asia. Wattanapisit *et al.* reported results from a study conducted among university students in seven South-East Asian countries in 2020–21. They examined co-occurrence of risks pertaining to physical activity, sedentary behaviour, sleep, substance use, diet, and mental well-being among study participants, using cluster analysis. Unlike us, they found little co-existence of risks, with risk factors occurring mostly in isolation. Also, women appeared to be at greater risk than men, particularly regarding sugary beverage consumption and poor mental well-being. These differences from our results, and the substantial inter-country differences they observed, reflect the role of local culture in shaping behavioural risks among youth [28]. Other studies among youth in South-East Asia have identified clusters of multiple behavioural risks for non-communicable disease, namely diet, physical activity, and alcohol use, pointing to some shared determinants of these risks [69–72]. These studies have also found co-occurrence of poor academic performance [28] and anxiety symptoms [70,72] with risk clusters. We did not examine these issues, and they could be important areas for future research. On the other hand, our inclusion of injury risks, suicide and sexual behaviour distinguishes our work from these studies.

Studies among adolescents and young adults in other settings, such as the United States and Europe, show some parallels with our results. Dietary and physical activity risks tend to be quite common across classes [73]. Even where classes differ in the prevalence of unhealthy diet, sedentary behaviour and insufficient physical activity, inter-class differences are not marked [74–76]. In contrast, substance use tends to show a much sharper variation across classes. The largest classes show low or no use of all substances, while a few small classes have very high prevalence for all substances [74,77]. Thus, our study, along with previous literature, shows that use of diverse substances such as alcohol, tobacco and cannabis tends to occur as highly concentrated syndemics.

## Strengths and limitations

One of the strengths of our study was the use of a multistage probability sampling approach, which allows our results to be generalised to the study population. Also, our use of a standard questionnaire covering a comprehensive range of behavioural risks among youth means that our study is programmatically relevant, and can be used for comparisons across settings and time. Though we removed some items and added local translations of a few terms, our findings are still internationally comparable, as the questionnaire was otherwise identical to the CDC YRBSS questionnaire. We believe that our identification of distinct high and low-risk classes will allow better targeting of preventive interventions.

One limitation of this study is the study size. While it was sufficient for estimating prevalence of common risks (prevalence >10%) and latent class modelling, it was not sufficient for estimating prevalence of rarer risks such as suicide attempts and unsafe sexual behaviours. In addition, we did not include out-of-college youth aged 18–22 years. While Chandigarh has one of the highest rates of college enrollment across the country [78], which means we missed out on relatively few individuals, these individuals still constitute an important population group with distinct health needs. Also, the high non-response rate reduced our study precision, and could introduce biases in our results if non-response was related

to the behavioural characteristics under investigation. The high rate of implausible data for some questions, especially sexual behaviour, reduced our study precision for these questions. This could have occurred because of inability of participants to comprehend the questions, or respondent fatigue, as these items were towards the end. The use of self-reported height and weight, rather than anthropometric measurements, could bias our BMI estimates downwards, because of social desirability for thinness in this population. Finally, extrapolation of our findings to youth less than 18 years of age or greater than 22 years of age should be done with caution.

**Implications of this study**

Our study provides a snapshot of avoidable health risks among college-attending youth in a prominent city in India. The questionnaire and methodology we used can be applied to conduct serial cross-sectional assessments, like the YRBSS in the United States, to conduct surveillance for behavioural risks among youth in India.

Our study also provides guidance for targeted health promotion among college students. While college or university-based interventions for reducing behavioural risks among youth have shown some effectiveness, we believe that their efficiency can be improved if they take interrelatedness of behaviours into account. For example, counsellors need to recognise that young individuals who use psychoactive substances may also be engaging in risky sex, and need help in both aspects. In the same way, those who face bullying or physical violence may also be at risk for driving-related injuries, and these risks should be addressed simultaneously. Since men have higher risk than women in most domains, they should be prioritised for risk-reduction interventions. For other risks such as insufficient physical activity, excessive screen time, inadequate sleep and unhealthy diets, which are widespread and do not show appreciable clustering, population-wide approaches may be preferable.

Future research should focus on reproducing these results across different parts of India, and developing context-specific interventions to reduce these risks. Another area of research could be tracking longitudinal changes in class membership over time [79].

## Supporting information

**S1 File. Study questionnaire.**
(DOCX)

**S2 File. Prevalences of different behavioural risks among college students aged 18–22 years in Chandigarh, with 95% confidence intervals.**
(DOCX)

**S3 File. Latent class model indices, for models with two to five latent classes.**
(DOCX)

**S4 File. Fitted prevalences of different behavioural risks among college students aged 18–22 years in Chandigarh, by latent class.**
(DOCX)

**S5 File. Cleaned dataset used for analysis, in comma-separated values (CSV) format.**
(CSV)

**S6 File. R code used to calculate overall prevalence and its 95% confidence intervals for behavioural risks.**
(DOCX)

**S7 File. MPlus code used to perform latent class analysis for behavioural risks.**
(DOCX)

## Acknowledgments

We thank the Department of Education (Chandigarh) for permission to conduct this study in colleges. We thank the principals of the colleges for their support, and the students for their participation in the study.

## Author contributions

**Conceptualization:** Vikas Kumar Bhatia, Shubh Mohan Singh, Pinnaka Venkata Maha Lakshmi.

**Data curation:** Vikas Kumar Bhatia.

**Formal analysis:** Vikas Kumar Bhatia, Adhish Kumar Sethi, Pinnaka Venkata Maha Lakshmi.

**Writing – original draft:** Vikas Kumar Bhatia, Adhish Kumar Sethi, Pratistha Sharma.

**Writing – review & editing:** Adhish Kumar Sethi, Pratistha Sharma, Shubh Mohan Singh, Pinnaka Venkata Maha Lakshmi.

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
