## [Decision Letter · Decision Letter 0]

6 Jun 2025

PONE-D-25-12897Do risky behaviours cluster among Indian youth? Novel insights from a latent class analysis.PLOS ONE?

Dear Dr. Lakshmi,

Thank you for submitting your manuscript to PLOS ONE. After careful consideration, we feel that it has merit but does not fully meet PLOS ONE’s publication criteria as it currently stands. Therefore, we invite you to submit a revised version of the manuscript that addresses the points raised during the review process.

Both reviewers identify some major issues. In addition to a point-by-point response to the issues raised, the study team needs to overall re-look at the ethical considerations and statistical analysis in light of the reviews. 

We look forward to receiving your revised manuscript.

Kind regards,

Yasir Alvi

Academic Editor

PLOS ONE

Journal Requirements:

2. Please remove all personal information, ensure that the data shared are in accordance with participant consent, and re-upload a fully anonymized data set.

Additional guidance on preparing raw data for publication can be found in our Data Policy (https://journals.plos.org/plosone/s/data-availability#loc-human-research-participant-data-and-other-sensitive-data) and in the following article: http://www.bmj.com/content/340/bmj.c181.long .

3. We note that there is identifying data in the Supporting Information file <PLOS One Supplementary File S4.csv>. Due to the inclusion of these potentially identifying data, we have removed this file from your file inventory. Prior to sharing human research participant data, authors should consult with an ethics committee to ensure data are shared in accordance with participant consent and all applicable local laws.

-Location data

4. We are unable to open your Supporting Information file PLOS One Supplementary File S5.R and PLOS One Supplementary File S6.R. Please kindly revise as necessary and re-upload.

Additional Editor Comments:

The author should strongly consider some of the following suggestion to improve the quality of the manuscript. 

**Title:**  The study population (college going students from Chandigarh) cannot be considered as Indian youth. Do revise the title and objective to reflect the actual study population. 

**Ethical considerations:**  There are a few critical issues here. The manuscript elicited information on suicide ideation/attempts, unprotected sex, substance use (including illicit drugs), and experiences of violence or victimization. WHO ethical guidelines calls for a priori plans to refer participants who need mental-health services, and provide counseling or resource sheets, which are missing here. 

In the research ethics frameworks, confidentiality has exceptions when a participant is at imminent risk of harm to self or others. Provide whether it was breached and did the findings of the study were shared with college administrations or concerned students to support targeted interventions?

Analysis: 

The manuscript statistical analysis had selected model based primarily on AIC, BIC, and interpretability of the fitted probabilities, they have not done any formal statistical tests, such as likelihood-ratio tests. While AIC and BIC are commonly used, the formal statistical tests can provide additional statistical evidence to support the chosen number of classes. The authors have not mention entropy value for the selected latent class model, which provides a key diagnostic for the quality of the latent class solution and the distinctiveness of the identified classes. Further having a sample size towards lower side and minimal differentiation of BIC curve between 4 and 5 classes, entropy values and likelihood-ratio tests are indeed indicated. 

The Latent class analysis results description may be improved by incorporating a key component including describing each class with its size and % of total sample, high and low endorsement of key variables, and its label. Try to describe each class using this for better clarity and allowing the reader to understand the characteristics of each identified class.

ie. Class 4 (n=xxx; 6.1% of the sample) was characterized by high endorsement of current substance use (p = xx) and unprotected sex (p = xx) but low probabilities on other behavioral risks; accordingly, we labeled it the ‘Substance Use & Sexual Risks’ class.”

Reviewers' comments:

Reviewer's Responses to Questions

**Comments to the Author**

1. Is the manuscript technically sound, and do the data support the conclusions?

Reviewer #1: Partly

Reviewer #2: Yes

2. Has the statistical analysis been performed appropriately and rigorously?

Reviewer #1: Yes

Reviewer #2: Yes

3. Have the authors made all data underlying the findings in their manuscript fully available?

Reviewer #1: Yes

Reviewer #2: Yes

4. Is the manuscript presented in an intelligible fashion and written in standard English?

Reviewer #1: Yes

Reviewer #2: Yes

Reviewer #1: Kindly mention, in detail, the details of division of colleges in different stratum and the basis adopted for it, since it has substantial impact on the generalizability of findings.

Also, was the data collection done anonymously, or some sort of identifier (like signed consent forms attached to questionnaire) was used. If anonymous, how was it maintained? Or if not anonymous, discuss the impact of social desirability bias in the limitations.

Also, the do mention the specific modifications done to the standard YRBSS questionnaire, and the YRBSS module you used should be referenced. How was the relevant modifications suitable in the Indian context and its impact on international comparability could also be discussed.

Reviewer #2: 

Line 1: The title should be according to journal guidelines. 

Line 2: The ‘period’ after the title should be deleted

Line 3: The short title should be in title case format.

Lines 40 & 41: Should be adjusted as earlier suggested

Lines 70-73: You can rephrase as ‘In 2019, the Global Burden of Disease study estimated that unintentional injuries and transport accidents together accounted for a quarter of 72 the deaths in the 10–24 years age group in India, with self-harm and violence accounting for 73 another 17% of the deaths in this age group.

Lines 86-87: “Similarly, a study in Bhubaneshwar found that almost half of college students engaged in physical activity ‘never’ or ‘occasionally’”. This statement is not clear. Kindly review for clarity.

Line 89: ‘Youth in India’ should be written as ‘Indian youth’

Line 98: ‘Youth in India’ should be written as ‘Indian youths’

Lines 99- 100: Should be rephrased as ‘Therefore, this study aimed to determine the prevalence of multiple behavioural risks among youths in a north Indian city, and identify clustering of risks within individuals, if any’.

Line 101: Since you did not investigate just one youth, then, the ‘youth’ should be written as ‘youths’.

Lines 103-106: ‘The city of Chandigarh is located in the northern part of India, and is notable for being a higher education hub, with a university and 26 colleges including over 80 departments.[28] Its student body comprises individuals hailing from northern India as well as other parts of the country.[29]’. This is describing the study location and should be moved to the ‘methods section’.

Lines 128-129: ‘This ratio was taken 129 keeping in mind the relative numbers of students in these colleges’. Kindly rephrase as ‘This proportion sampling technique was employed, given the relative student population in these colleges’.

Lines 147-148: ‘One investigator’ cannot continue as ‘they’ in the following sentence. Kindly review for clarity.

Lines 173-173: ‘We defined ‘insufficient physical activity’ as being 174 physically active for at least 60 minutes a day for less than five days in the last week’. Please review this as it may be correct. Someone who had exercised for many hours for 4 days in a week cannot be considered inactive.

Line 270: ‘Are like’ should be rephrased as ‘were similar to those of….’

Line 272: Add respectively after this: ‘17% for 20–29 years.[11]’.

Line 273: You can consider changing the verb ‘are’ to ‘may’ as your study design, a cross-sectional study and sample size, is not strong enough to make a national inference about Indian youths.

Line 313: Please can you provide the age range of these young men for clearer comparison with your study participants.

Line 365: ‘Youth’ should be ‘youths’

Line 375: ‘means’ should be ‘meant’.

Line 376: ‘is’ should be ‘was’.

Line 379: Since you are reporting a limitation of a study that was already conducted, I think it should be reported in past tense; ‘is’ should be ‘was’

Line 380: ‘has’ should be ‘had’

Line 386: Kindly change ‘happened’ to ‘occurred’.

Line 388: Change ‘is’ to ‘was’.

Line 391: Change ‘youth’ to ‘youths’.

**Do you want your identity to be public for this peer review?** For information about this choice, including consent withdrawal, please see our Privacy Policy

Reviewer #1: No

Reviewer #2: No

---

## [Author Response · Author response to Decision Letter 1]

26 Jun 2025

Editorial comments

Authors’ response:

We have now formatted the files and changed the file names, in accordance with the journal style.

• We have added page numbers at the bottom of each page in the main file.

• We have provided author names as a byline, with affiliations indicated by numbers in superscript.

• We have added heading levels (Heading 1 and Heading 2) as per the journal style guide.

• We have labelled figures as ‘Fig’, and named the figure files as ‘Fig1.tiff’, ‘Fig2.tiff’, and ‘Fig3.tiff’.

• We have named the supporting information files as ‘S1 File’, ‘S2 File’, so on.

2. Please remove all personal information, ensure that the data shared are in accordance with participant consent, and re-upload a fully anonymized data set.

Authors’ response:

The dataset we have uploaded does not contain any direct personal identifiers. It is anonymised, and does not contain the name or initials of any participant. We have now removed the variables ‘educational stream’ and ‘college year’ from the dataset. It is not possible to use the information remaining in our dataset to uniquely identify any individual. Our data sharing is in accordance with participant consent.

3. We note that there is identifying data in the Supporting Information file <PLOS One Supplementary File S4.csv>. Due to the inclusion of these potentially identifying data, we have removed this file from your file inventory. Prior to sharing human research participant data, authors should consult with an ethics committee to ensure data are shared in accordance with participant consent and all applicable local laws.

-Location data

Authors’ response:

We are committed to maintaining the privacy of study participants. The dataset we have uploaded does not contain any direct personal identifiers. It does not contain the name, initials, address or college name for any participant.

Also, we have now removed the variables ‘educational stream’ and ‘college year’ from the dataset (S5 File). These were indirect identifiers, and did not contribute to the main analysis. We have modified the Results accordingly.

P 10 lines 235–236

‘Among the 752 respondents, 268 (35.6%) identified as male, 484 (64.4%) identified as female, and none as transgender. The median age of the respondents was 19 years (IQR 19 to 21 years).’

Now, only two indirect identifiers remain in our dataset: age and gender. It is not possible to use the combined information from these two variables to uniquely identify any individual. These variables contributed to our results, and without them, readers may not be able to reproduce our findings.

Regarding the serial number column, we added it after entering the data. The serial number cannot be used to identify any individual.

We reiterate that our data sharing is in accordance with the protocol which our institute’s ethics committee approved. It is also in accordance with participant consent.

4. We are unable to open your Supporting Information file PLOS One Supplementary File S5.R and PLOS One Supplementary File S6.R. Please kindly revise as necessary and re-upload.

Authors’ response: We had earlier uploaded these files as R scripts. We are now uploading the files in Microsoft Word format, as ‘S6 File.docx’ and ‘S7 File.docx’.

S6 File: R code used to calculate overall prevalence and its 95% confidence intervals for behavioural risks

S7 File: MPlus code used to perform latent class analysis for behavioural risks

Additional editor comments

The author should strongly consider some of the following suggestion to improve the quality of the manuscript.

1. Title: The study population (college going students from Chandigarh) cannot be considered as Indian youth. Do revise the title and objective to reflect the actual study population.

Authors’ response: We agree. We have now modified the title to, ‘Do risky behaviours cluster among college students in Chandigarh, India? Novel insights from a latent class analysis’. We have also changed the short title to ‘Latent class analysis of behavioural risks among Chandigarh youth’.

We had already included the study population in the objectives as follows:

P 4 lines 93–94

‘Therefore, we planned this study to determine the prevalence of multiple behavioural risks among youth in a north Indian city, and identify clustering of risks within individuals, if any.’

2. Ethical considerations: There are a few critical issues here. The manuscript elicited information on suicide ideation/attempts, unprotected sex, substance use (including illicit drugs), and experiences of violence or victimization. WHO ethical guidelines calls for a priori plans to refer participants who need mental-health services, and provide counseling or resource sheets, which are missing here.

In the research ethics frameworks, confidentiality has exceptions when a participant is at imminent risk of harm to self or others. Provide whether it was breached and did the findings of the study were shared with college administrations or concerned students to support targeted interventions?

Authors’ response: We appreciate the editor’s attention to this very important point. Our study team had discussed this while planning the study. While collecting data with direct identifiers (such as names) would have allowed linkage with care, it would also have created concerns about confidentiality. It might also have influenced participants’ responses due to social desirability. Considering these factors, we decided to have anonymous data collection. We had mentioned this in the proposal that we submitted to the institute ethics committee, before starting the study. The institute ethics committee approved the proposal.

In terms of maintenance of anonymity, our methodology is the same as the original methodology used by the US CDC. See, for example:

Brener ND, Kann L, Shanklin S, Kinchen S, Eaton DK, Hawkins J, et al. Methodology of the Youth Risk Behavior Surveillance System — 2013. Morb Mortal Wkly Rep Recomm Rep. 2013;62: 1–23.

Underwood JM, Brener N, Thornton J, Harris WA, Bryan LN, Shanklin SL, et al. Overview and Methods for the Youth Risk Behavior Surveillance System - United States, 2019. MMWR Suppl. 2020 Aug 21;69(1):1-10. doi: 10.15585/mmwr.su6901a1.

We have now clarified these issues in the manuscript.

Study tool, p 7, lines 159–163

‘In total, the questionnaire comprised 74 questions. To maintain anonymity, it did not ask for the participant’s name or address. Most of the questions involved selecting responses from given options. Only the height and weight questions involved writing numerical digits in the data collection form. Hence, it was not possible to identify students by their handwriting.’

Ethical considerations, p 9, lines 216–219

‘Though the consent forms bore the signatures and names of participants, it was not possible to link them with the data collection forms which the participants submitted, thus ensuring anonymity. Given the anonymous nature of data collection, it was not possible for us to link participants with care if they reported behavioural risks (e.g., suicidal ideation or substance use).’

3. Analysis: The manuscript statistical analysis had selected model based primarily on AIC, BIC, and interpretability of the fitted probabilities, they have not done any formal statistical tests, such as likelihood-ratio tests. While AIC and BIC are commonly used, the formal statistical tests can provide additional statistical evidence to support the chosen number of classes. The authors have not mention entropy value for the selected latent class model, which provides a key diagnostic for the quality of the latent class solution and the distinctiveness of the identified classes. Further having a sample size towards lower side and minimal differentiation of BIC curve between 4 and 5 classes, entropy values and likelihood-ratio tests are indeed indicated.

Authors’ response: Thank you for this excellent suggestion. We have now revised the analysis. Since we could not obtain the entropy and likelihood ratio test p-values in R, we have now used MPlus for the latent class analysis.

Abstract, p 2

Lines 34–35: ‘We selected the most appropriate latent class model based on fitted probabilities, likelihood ratio tests, entropy, Akaike information criterion and Bayesian information criterion.’

Lines 37–38: ‘Latent class analysis identified four classes: multiple risks (8.8%), substance use and sexual risks (5.4%), low-risk (73.7%), and victimisation and injury risks (12.1%).’

Statistical analysis and reporting of results

p 8, lines 176–178

‘We entered the data into a spreadsheet, and then imported the spreadsheet to statistical analysis software. We did data processing, cleaning and prevalence estimation with R 4.3.0 (R Foundation for Statistical Computing, Vienna, Austria).’

P 9, lines 202–208

‘To identify clustering of risk factors, we performed latent class analysis using MPlus Version 8.4 (Muthen & Muthen). We included the 27 behavioural risks as binary variables, with gender as a covariate. We ran models with two to seven latent classes. We examined the fitted probabilities, entropy, p-values from the Vuong–Lo–Mendell–Rubin likelihood ratio test, the Akaike information criterion (AIC), Bayesian information criterion (BIC) and sample size-adjusted Bayesian information criterion (aBIC) for each model, to select the model with best performance.’

Results, p 11, lines 269–275

‘We selected the model with four classes as it was able to distinguish meaningful classes, had a relatively high entropy (0.916), and reasonably low AIC, BIC and aBIC (Figure 2 and S3 File). Also, the Vuong–Lo–Mendell–Rubin likelihood ratio test showed substantial improvement in fit up to four classes, but not on adding a fifth class (p=0.366 for five versus four classes). Figure 3 and S4 File show the pattern of item probabilities within these four classes. Some of the risks, namely nutritional risks, digital device use and sleep, did not differ appreciably across the classes. The other risks showed a distinctive pattern across classes.’

Figure 2: Assessment of latent class models for behavioural risks among youth in Chandigarh. Model fit was assessed using a. AIC, BIC and aBIC, and b. entropy, for models with different numbers of classes.

We have also added numerical values of these indices as supporting information:

S3 File: Latent class model indices, for models with two to five latent classes

We have also revised Table 1, Figure 3 and S4 File accordingly.

4. The Latent class analysis results description may be improved by incorporating a key component including describing each class with its size and % of total sample, high and low endorsement of key variables, and its label. Try to describe each class using this for better clarity and allowing the reader to understand the characteristics of each identified class.

ie. Class 4 (n=xxx; 6.1% of the sample) was characterized by high endorsement of current substance use (p = xx) and unprotected sex (p = xx) but low probabilities on other behavioral risks; accordingly, we labeled it the ‘Substance Use & Sexual Risks’ class.”

Authors’ response: We have taken note of this suggestion. For clarity, we have now provided a detailed description of the latent classes in the Results.

P 12, lines 287–305

“Class 1 (estimated to constitute 8.8% of the population) was characterised by high endorsement of risks in almost all domains. These included driving under influence of alcohol (48.2%), mobile use while driving (74.4%), involvement in physical fights (51.1%), depressive symptoms (41.5%), cigarette smoking (52.1%), smokeless tobacco use (12.4%), alcohol use (76.8%), cannabis use (40.3%), substance use before last sexual intercourse (26.9%), and condomless sex (26.6%). Hence, we labelled this class as ‘multiple risks’.

Class 2 (5.4%) was marked by high endorsement of current cigarette smoking (46.8%), current alcohol use (100.0%), binge drinking (100.0%), four or more sexual partners in lifetime (11.0%), substance use before last sexual intercourse (10.9%), and condomless sex (14.6%). This class had low estimated prevalence of driving-related risks, physical violence and bully-victimisation. Accordingly, we labelled it as ‘substance use and sexual risks’.

In contrast, individuals in Class 3 (73.7%) had low estimated risk in most domains. Their prominent risks were diet, physical activity, digital device use, and sleep. Since these were uniformly high across almost all classes, we labelled Class 3 as ‘low risk’.

Finally, Class 4 (12.1%) was distinguished by high estimated prevalence of mobile use while driving (21.6%), involvement in physical fights (25.0%), bully-victimisation on campus (13.5%), and cyber bully-victimisation (21.6%). This was the only class with low prevalence of insufficient physical activity (0.0%) and inadequate muscle strengthening exercises (0.0%). We labelled this class as ‘victimisation and injury risks’.”

Reviewers' comments:

Reviewer #1

1. Kindly mention, in detail, the details of division of colleges in different stratum and the basis adopted for it, since it has substantial impact on the generalizability of findings.

Authors’ response: We thank the reviewer for this constructive comment.

At the time we conducted this study, Chandigarh had women’s colleges and co-educational colleges (but not a “men-only” college). From data available at that time, we had estimated that about one-sixth of the student population was enrolled in women’s colleges. Hence, we randomly selected one women’s college and five co-educational colleges. We have mentioned this in the manuscript:

P 6, lines 128–130:

‘Out of 32 government and private colleges in Chandigarh, six colleges were randomly selected—one women’s college and five co-educational colleges. This ratio was taken keeping in mind the relative numbers of students in these

---

## [Decision Letter · Decision Letter 1]

3 Sep 2025

PONE-D-25-12897R1Do risky behaviours cluster among college students in Chandigarh, India? Novel insights from a latent class analysisPLOS ONE?

Dear Dr. Lakshmi,

Thank you for submitting your manuscript to PLOS ONE. After careful consideration, we feel that it has merit but does not fully meet PLOS ONE’s publication criteria as it currently stands. Therefore, we invite you to submit a revised version of the manuscript that addresses the points raised during the review process.

**ACADEMIC EDITOR:** Your study addresses an important issue. Please clarify the definitions of risky behaviours, justify methodological choices, and refine interpretation of latent classes, particularly for low-prevalence behaviours and gender differences. 

We look forward to receiving your revised manuscript.

Kind regards,

Bijit Biswas, MBBS, MD, DNB

Academic Editor

PLOS ONE

Journal Requirements:

Additional Editor Comments:

Your study addresses an important issue. Please clarify the definitions of risky behaviours, justify methodological choices, and refine interpretation of latent classes, particularly for low-prevalence behaviours and gender differences.

Reviewers' comments:

Reviewer's Responses to Questions

**Comments to the Author**

Reviewer #2: All comments have been addressed

Reviewer #3: (No Response)

Reviewer #4: All comments have been addressed

Reviewer #5: (No Response)

2. Is the manuscript technically sound, and do the data support the conclusions?

Reviewer #2: Yes

Reviewer #3: Yes

Reviewer #4: Partly

Reviewer #5: Yes

3. Has the statistical analysis been performed appropriately and rigorously?

Reviewer #2: Yes

Reviewer #3: Yes

Reviewer #4: Yes

Reviewer #5: Yes

4. Have the authors made all data underlying the findings in their manuscript fully available?

Reviewer #2: Yes

Reviewer #3: Yes

Reviewer #4: Yes

Reviewer #5: Yes

5. Is the manuscript presented in an intelligible fashion and written in standard English?

Reviewer #2: Yes

Reviewer #3: Yes

Reviewer #4: Yes

Reviewer #5: Yes

Reviewer #2: The author attended to the comments raised. The editor made a comprehensive review of the manuscript which made it more interesting and professional. Please note that extrapolation in science should always be done with caution.

Reviewer #3: Thank you for the opportunity to review the article “Do risky behaviours cluster among college students in Chandigarh, India? Novel insights from a latent class analysis”. This novel article addresses the prevalence of behavioral risks and identifies their clustering among young adults in Chandigarh, India.

Here are some suggestions for the authors:

Regarding the formatting of the manuscript, I would adhere to traditional sections commonly found in a standard scientific manuscript, particularly in the method section, including participants, procedure, and statistical analysis. This way, the text is easier to follow.

The data from lines 194-216 may be more visually appealing and understandable when displayed in a table.

Overall, I believe the manuscript is well-written and addresses an interesting topic. Congratulations to the authors for their work.

Reviewer #4: Strengths

• This study makes an important contribution by applying Latent Class Analysis (LCA) to explore risky behaviours among college students in Northern India. Given the limited data from South Asia, the findings are valuable for both public health and education policy.

Major Comments

1. Class labelling (Lines 217, Supplementary File S4)

o The naming of latent classes could be reconsidered. For example, the “low risk” class still shows risks in nutrition and physical activity. Also, Class 1 appears to have higher sexual risk percentages than Class 2, which makes the current label “substance use and sexual risks” somewhat confusing. A clearer rationale or adjusted labels would improve interpretation.

2. Discussion alignment with the title (Lines 255, Discussion)

o Since the title emphasizes clustering through LCA, the Discussion would be stronger if it more clearly highlighted insights from the LCA results, with other findings used as supportive points.

3. Low-prevalence behaviours (S2 File)

o Behaviours such as sexual activity (3%) and suicide attempt (0.4%) have very low prevalence in this dataset. Discussion of these findings may need to be more cautious, or kept brief, to avoid overstating their implications.

4. Gender differences (Lines 345–349)

o The claim that men engage in more risky behaviours due to biological and social factors is interesting, but no supporting data were presented in this study. It may be better to frame this point as a possible future research direction rather than as a conclusion.

Reviewer #5: To provide context to the authors, I am a new reviewer brought on to review the manuscript between reviews. I have read both the responses to the editor and previous reviewers prior to reading the manuscript. The authors have addressed the reviewers’ and editor’s comments thoroughly. However, I believe there are still some issues in the reported analysis that require clarification.

Introduction: The introduction needs to be strengthened for better logical flow and coherence

1. In this manuscript, risky behavior is a key concept but its definition is unclear. The introduction lists different sets of behaviors in the first and second paragraphs, which are inconsistent and lack clear logic. For example, tobacco use and unhealthy diets appear once but are not discussed further, while it is questionable whether transport accidents should be classified as risky behaviors. The authors should provide a clear and consistent definition of risky behavior and ensure that the examples cited are logically connected and evidence-based.

2. Sexual risk behavior plays an important role in the results, where Class 2 is labeled as substance use and sexual risks, and it is also extensively discussed in the discussion section. However, the introduction provides very limited description of sexual risk behavior. Moreover, there is a substantial body of literature on sexual risk among college students, yet none of these studies are cited. The introduction would be strengthened by a clearer discussion of sexual risk behavior and by incorporating relevant evidence from existing research.

3.. The study focuses on college students, yet the introduction primarily discusses youth, and only mentions college students in the final paragraph. This issue was raised in previous rounds of review, but it does not appear to have been adequately addressed. The rationale for focusing specifically on college students is not sufficiently convincing. For instance, the authors suggest that college students have increased autonomy because they are less controlled by parents, but this characteristic may also apply to young people who do not attend college. The introduction would be strengthened by providing evidence that risk behaviors are particularly prevalent among college students compared to their non-college peers.

Methods

4. Regarding the sample size calculation, it is unclear why the authors used smokeless tobacco use as the reference. This study includes a range of risky behaviors, not just tobacco use. It may be more appropriate to base the calculation on the prevalence of risky behaviors among college students reported in previous studies, for example using the median rate across different behaviors.

5. The inclusion criteria restrict participants to ages 18–22, but the rationale for this age range is unclear. In the introduction, the cited evidence refers to individuals under 24. Are college students in India generally under 22? If some students are older due to gaps or delayed enrollment (e.g., 22–24 years), will they be excluded? The authors should clarify the reason for this age restriction and justify why older college students are not included.

6. line 166, the authors mention that they listed 27 important behavioral risks. It is unclear what the basis for this selection is. Are these all behavioral risks included in the YRBSS, or were they selected by the authors as deemed important? If the latter, a supporting reference should be provided. If the YRBSS includes only these 27 risks, this should be clearly stated in the Study Tool section.

7. Supplementary Table S2 shows that the 27 behavioral risks are grouped into seven categories: Injury risks, Victimization, Depression and suicide risk, Substance use, Nutrition and diet, Physical activity, sedentary behavior and sleep, and Sexual behavior. The authors should describe this categorization in the Study Tool section. Without this explanation, readers may find it difficult to understand the composition and classification of the 27 important behavioral risks.

Results

8. The label for Class 2 (substance use and sexual risks) may be misleading. As shown in Supplementary File S4, Class 2 is primarily characterized by alcohol use and smoking, whereas cannabis use and tobacco use are both zero. In fact, cannabis and tobacco use are significantly higher in the Multiple Risk group. Naming Class 2 as substance use and sexual risks could therefore give a misleading impression of its behavioral profile.

9. In Class 2, 10.9% of participants reported substance use before their last intercourse, yet the data show that cannabis use and tobacco use are both zero. This appears inconsistent. If substance use before sexual activity does not include cannabis and tobacco, the assessment of substance use may be incomplete, potentially overlooking important behaviors.

**Do you want your identity to be public for this peer review?** For information about this choice, including consent withdrawal, please see our Privacy Policy

Reviewer #2: No

Reviewer #3: No

Reviewer #4: No

Reviewer #5: No

---

## [Author Response · Author response to Decision Letter 2]

24 Sep 2025

Editorial comments

Your study addresses an important issue. Please clarify the definitions of risky behaviours, justify methodological choices, and refine interpretation of latent classes, particularly for low-prevalence behaviours and gender differences.

Authors’ response

Thank you for your comments. We have now added a definition of ‘risky behaviours’ or ‘behavioural risks’ in the Introduction (p 4 lines 47–49). We have revised the Introduction and Methods section, to clarify our methods and our rationale for them. We have also renamed two of the latent classes, to better reflect their behavioural patterns. Class 2 is now labelled, ‘smoking, alcohol use and sexual risks’, while Class 3 is labelled, ‘only dietary and physical activity risks’. We have shortened our discussion of low-prevalence behaviours, and increased coverage of latent class analysis in the discussion. We have also toned down our conclusions on the mechanisms of gender differences, presenting them as future research directions instead.

Reviewers' comments:

Reviewer #2: The author attended to the comments raised. The editor made a comprehensive review of the manuscript which made it more interesting and professional. Please note that extrapolation in science should always be done with caution.

Authors’ response

We thank the reviewer for their inputs. We agree that extrapolation should be done cautiously. While drawing inferences from our findings, we have been cognisant of the limitations of our study (p 23 lines 452–466).

Reviewer #3: Thank you for the opportunity to review the article “Do risky behaviours cluster among college students in Chandigarh, India? Novel insights from a latent class analysis”. This novel article addresses the prevalence of behavioral risks and identifies their clustering among young adults in Chandigarh, India. Here are some suggestions for the authors:

Regarding the formatting of the manuscript, I would adhere to traditional sections commonly found in a standard scientific manuscript, particularly in the method section, including participants, procedure, and statistical analysis. This way, the text is easier to follow.

Authors’ response

We thank the reviewer for this suggestion. We agree that this could be another way of organising the manuscript. In line with journal guidelines, we have arranged our manuscript under the headings ‘Introduction’, ‘Methods’, ‘Results’ and ‘Discussion’. To maintain the flow of the Methods section, we have organised the Methods section in the following sub-headings.

Study setting and design

Study population and eligibility criteria

Study size and sampling strategy

Study tool

Data collection

Statistical analysis and reporting of results

Ethical considerations

The data from lines 194-216 may be more visually appealing and understandable when displayed in a table.

Authors’ response

We agree with the reviewer. Hence, we have presented the prevalence findings as a table in S2 File. To make the findings visually appealing and engaging, we have also presented them graphically in Figure 1.

Overall, I believe the manuscript is well-written and addresses an interesting topic. Congratulations to the authors for their work.

Authors’ response

Thank you very much for your interest in our work. We are happy to receive your helpful comments and appreciation.

Reviewer #4: Strengths

• This study makes an important contribution by applying Latent Class Analysis (LCA) to explore risky behaviours among college students in Northern India. Given the limited data from South Asia, the findings are valuable for both public health and education policy.

Authors’ response

Thank you for your positive assessment. We hope that our work will make a useful contribution to public health and education policy in South Asia and elsewhere.

Major Comments

1. Class labelling (Lines 217, Supplementary File S4)

o The naming of latent classes could be reconsidered. For example, the “low risk” class still shows risks in nutrition and physical activity. Also, Class 1 appears to have higher sexual risk percentages than Class 2, which makes the current label “substance use and sexual risks” somewhat confusing. A clearer rationale or adjusted labels would improve interpretation.

Authors’ response

We agree with the reviewer’s observation. We believe that renaming class 3 to ‘only dietary and physical activity risks’ will mitigate much of the confusion. So, we have made the following changes.

Abstract, lines 34–36:

‘Latent class analysis identified four classes: multiple risks (8.8%), smoking, alcohol use and sexual risks (5.4%), only dietary and physical activity risks (73.7%), and victimisation and injury risks (12.1%).’

Results, pp 14–15, lines 274–275

“Hence, we labelled Class 3 as ‘only dietary and physical activity risks’.”

We have also updated the class labels in Table 1 and S4 File.

Similarly, we have renamed Class 2 as ‘smoking, alcohol use and sexual risks’.

We agree that Class 1 has higher prevalence of sexual risks than Class 2. It is also important to recognise that, in any latent class analysis, the class names are short labels. They are meant for readers to quickly pick up patterns. The brevity of these labels means that some information will invariably be left out, and they cannot provide a comprehensive description of all the risk factors and their prevalences. This is why we have provided detailed numbers in the Results and S4 file, and a graph in Figure 3. We think that, with these details and the revised class labels, the readers will be able to get an in-depth interpretation of the patterns in the latent classes.

2. Discussion alignment with the title (Lines 255, Discussion)

o Since the title emphasizes clustering through LCA, the Discussion would be stronger if it more clearly highlighted insights from the LCA results, with other findings used as supportive points.

Authors’ response

We thank the reviewer for this suggestion. We have now greatly shortened the discussion on individual risk factors (pp 16–20). We have expanded the discussion on latent class analysis as follows.

P 20 lines 388–392

‘Similarly, the co-occurrence of victimisation, depression, and suicidal ideation, as seen in Class 1 in our study, has been noted in earlier work [62]. Indeed, these constructs might be causally linked, as shown by previous studies. For example, longitudinal studies show that cyber-bullying victimisation among adolescents correlates with subsequent depressive symptoms [63,64].’

P 22 lines 433–441

‘Studies among adolescents and young adults in other settings, such as the United States and Europe, show some parallels with our results. Dietary and physical activity risks tend to be quite common across classes [73]. Even where classes differ in the prevalence of unhealthy diet, sedentary behaviour and insufficient physical activity, inter-class differences are not marked [74–76] In contrast, substance use tends to show a much sharper variation across classes. The largest classes show low or no use of all substances, while a few small classes have very high prevalence for all substances [74,77]. Thus, our study, along with previous literature, shows that use of diverse substances such as alcohol, tobacco and cannabis tends to occur as highly concentrated syndemics.’

P 24 line 484–5

‘Another area of research could be tracking longitudinal changes in class membership over time [79].’

3. Low-prevalence behaviours (S2 File)

o Behaviours such as sexual activity (3%) and suicide attempt (0.4%) have very low prevalence in this dataset. Discussion of these findings may need to be more cautious, or kept brief, to avoid overstating their implications.

Authors’ response

We agree with the reviewer’s suggestion. We have now substantially shortened the discussion on sexual behaviours (p 18 lines 335–361) and suicide (pp 19–20 lines 363–378).

4. Gender differences (Lines 345–349)

o The claim that men engage in more risky behaviours due to biological and social factors is interesting, but no supporting data were presented in this study. It may be better to frame this point as a possible future research direction rather than as a conclusion.

Authors’ response

We agree with the reviewer on this point. While we have shown gender differences in risk clustering (Table 1), we have not demonstrated any biological or social mechanism for this difference. Accordingly, we have now toned down our conclusions in the Discussion. We have now presented them as possible explanations based on prior literature, and directions for future research.

Pp 20–21, lines 393–399

‘We observed a greater tendency of men to engage in risky behaviour. While we did not explore mechanisms for this difference, previous work shows that biological differences between men and women could play a role. The male hormone testosterone is known to promote aggression and risk-taking.[65] This might be reinforced by social norms which validate risky endeavours by men and discourage such endeavours by women.[66] Uncovering the relative contributions of nature and nurture to gendered risk-taking could be an idea for future research in India.’

Reviewer #5: To provide context to the authors, I am a new reviewer brought on to review the manuscript between reviews. I have read both the responses to the editor and previous reviewers prior to reading the manuscript. The authors have addressed the reviewers’ and editor’s comments thoroughly. However, I believe there are still some issues in the reported analysis that require clarification.

Introduction: The introduction needs to be strengthened for better logical flow and coherence

Authors’ response

We have tried to improve the flow of the Introduction based on the reviewer’s suggestions.

1. In this manuscript, risky behavior is a key concept but its definition is unclear. The introduction lists different sets of behaviors in the first and second paragraphs, which are inconsistent and lack clear logic. For example, tobacco use and unhealthy diets appear once but are not discussed further, while it is questionable whether transport accidents should be classified as risky behaviors. The authors should provide a clear and consistent definition of risky behavior and ensure that the examples cited are logically connected and evidence-based.

Authors’ response

We agree that defining risky behaviours is important for this study. Hence, we have now included the definition in the Introduction.

P 4 lines 47–52

‘While definitions vary, ‘risky behaviours’ or ‘behavioural risks’ are commonly conceptualised as avoidable actions, or omissions, of individuals which increase chances of adverse health outcomes for themselves or others [4]. Some examples of behavioural risks which often begin in youth are unsafe driving, tobacco use, alcohol use, drugs, violence, unsafe sexual behaviours and unhealthy diets.[5]’

We have also expanded the introduction to show more clearly the links of risky behaviours, including non-use of seat belts, tobacco use, and unhealthy diets, with health outcomes.

P 4 lines 59–64

‘In that year, alcohol use was found to be the most important behavioural risk for death in the 10–24 years age group, accounting for 2.6% of total deaths. Similarly, unsafe sex accounted for 0.9% of deaths and tobacco use for 0.3% of deaths.[7] Many of the deaths in transport accidents in India are attributable to behaviours such as non-use of seat belts and helmets [8]. Unhealthy diets and insufficient physical activity, when established during youth and persistent in the long term, show their effects late in the life course, in the form of non-communicable diseases like cardiovascular disease [9].’

2. Sexual risk behavior plays an important role in the results, where Class 2 is labeled as substance use and sexual risks, and it is also extensively discussed in the discussion section. However, the introduction provides very limited description of sexual risk behavior. Moreover, there is a substantial body of literature on sexual risk among college students, yet none of these studies are cited. The introduction would be strengthened by a clearer discussion of sexual risk behavior and by incorporating relevant evidence from existing research.

Authors’ response

We thank the reviewer for highlighting this. In the Introduction section, we have now elaborated on sexual risks among youth in India, based on a more extensive review of literature.

P 5 lines 76–84

‘Sexual activity in youth is rare in India, as compared to other countries [17]. Yet, among those who are sexually active, the prevalence of condomless sex may be quite high. This is particularly true for boys and young men. A study conducted in Chandigarh in 2020 found that nearly two-thirds of sexually active college students had unprotected sex [18]. Another study among men aged 18–24 in Ballabgarh, Haryana, found that almost one-third of those sexually active had engaged in condomless sex [10]. Use of psychoactive substances before sex is another concern, as it impairs judgement and decreases condom use [19]. There are limited data on this practice among youth in India [20].’

3. The study focuses on college students, yet the introduction primarily discusses youth, and only mentions college students in the final paragraph. This issue was raised in previous rounds of review, but it does not appear to have been adequately addressed. The rationale for focusing specifically on college students is not sufficiently convincing. For instance, the authors suggest that college students have increased autonomy because they are less controlled by parents, but this characteristic may also apply to young people who do not attend college. The introduction would be strengthened by providing evidence that risk behaviors are particularly prevalent among college students compared to their non-college peers.

Authors’ response

We thank the reviewer for bringing this to our attention. We have revised the Introduction accordingly. We have now elaborated on our rationale for including college students in this study.

P 6 lines 98–105

‘We focused on youth attending colleges for higher education. We did this for several reasons. One, colleges were sites where we could readily access eligible individuals, using the limited resources that we had for this study. Two, colleges are an important avenue for youth health promotion programmes [33,34]. Three, in the Indian context, higher education is a critical phase where the increased autonomy from parental control means that a person is likely to adopt unsafe behaviours.[35,36] College students may show greater prevalence of behavioural risks, such as binge drinking, than their non-college peers [37].’

Methods

4. Regarding the sample size calculation, it is unclear why the authors used smokeless tobacco use as the reference. This study includes a range of risky behaviors, not just tobacco use. It may be more appropriate to base the calculation on the prevalence of risky behaviors among college students reported in previous studies, for example using the median rate across different behaviors.

Authors’ response

We thank the reviewer for taking note of this. For estimating a prevalence p with relative error r and confidence level 100(1−α)%, the required sample size is (Z^2)p(1−p)/((rp)^2), where Z is the standard normal deviate corresponding to the chosen α. It can be shown that, for a constant r, this sample size decreases monotonically with increasing p. In other words, for a given relative error, outcomes with lower prevalence (i.e., rare outcomes) need greater sample size for prevalence estimation. We chose smokeless tobacco use as its anticipated prevalence, 10.8%, was on the lower side among the behaviours we were interested in. With this prevalence and 20% relative error, we got a sample size of 793.

An interesting question would be, is this sample size sufficient for the other risks too? With this sample size of 793, we estimated reasonable relative errors for common behavioural risks. For example, with this sample size, for overweight (anticipated prevalence 31.1%), we would have a relative error of 10.4%. On the other hand, for the rarer risks, we anticipated worse precision. For example, for condomless sex (anticipated prevalence 2.6%), we anticipated t

---

## [Decision Letter · Decision Letter 2]

11 Nov 2025

PONE-D-25-12897R2Do risky behaviours cluster among college students in Chandigarh, India? Novel insights from a latent class analysisPLOS ONE?

Dear Dr. Lakshmi,

Thank you for submitting your manuscript to PLOS ONE. After careful consideration, we feel that it has merit but does not fully meet PLOS ONE’s publication criteria as it currently stands. Therefore, we invite you to submit a revised version of the manuscript that addresses the points raised during the review process.

We look forward to receiving your revised manuscript.

Kind regards,

Yuan-Pang Wang, M.D., Ph.D.

Academic Editor

PLOS ONE

Journal Requirements:

Additional Editor Comments:

One of reviewers requested additional work for the label of class and consistent use of terminology. Please revise.

Reviewer's Responses to Questions

**Comments to the Author**

Reviewer #4: (No Response)

Reviewer #5: All comments have been addressed

2. Is the manuscript technically sound, and do the data support the conclusions?

Reviewer #4: Yes

Reviewer #5: Yes

3. Has the statistical analysis been performed appropriately and rigorously?

Reviewer #4: Yes

Reviewer #5: Yes

4. Have the authors made all data underlying the findings in their manuscript fully available?

Reviewer #4: Yes

Reviewer #5: Yes

5. Is the manuscript presented in an intelligible fashion and written in standard English?

Reviewer #4: Yes

Reviewer #5: Yes

Reviewer #4: Thank you for submitting the revised version of your manuscript. The study remains important in the context of public health and education policy, and the analytical approach is appropriate. The structure and clarity have improved compared with the previous version. However, one previously raised issue has not yet been adequately addressed, and one new point requires clarification and revision.

1. Class Labelling (Lines 217; Supplementary File S4)

(Previously raised in Round 1 but not sufficiently resolved)

Class 2 continues to be labelled “smoking, alcohol use and sexual risks,” which remains inconsistent with both the statistical results and conceptual interpretation.

a. Very low prevalence of sexual behaviour variables

According to the S2 File, variables related to sexual behaviour have extremely low prevalence, with the highest frequency being 21 participants (about 3%). Such a small subsample cannot provide stable class estimation; low-prevalence indicators often yield unstable parameter estimates in LCA and therefore should not define a class.

b. Fitted prevalence patterns do not support the label

The S4 File shows that sexual-risk indicators are actually higher in Class 1 (multiple risks) than in Class 2. For example:

- Condom not used at last intercourse: Class 1 = 26.6%, Class 2 = 14.6%

- Four or more sexual partners in lifetime: Class 1 = 16.1%, Class 2 = 11.0%

These results indicate that sexual-risk behaviours are characteristic of Class 1, not Class 2. Including “sexual risks” in the Class 2 label may therefore be misleading.

Recommendation:

Please reconsider the naming of Class 2 to reflect its dominant behavioural profile more accurately—for example:

- “smoking and alcohol-related risks”, or

- “substance use–dominant risks.”

Adding a brief statement in the Methods or Results section explaining the rationale for final class names (e.g., theoretical justification, indicator strength, or item-response probabilities) would also improve transparency.

2. Terminology Consistency

(New observation in Round 2)

The manuscript alternates between “risky behaviours” and “behavioural risks.” Although the two expressions are related, consistent terminology is necessary for clarity and precision.

Given that this study adopts an epidemiological and risk-modelling perspective, “behavioural risks” is the more technically accurate term because it frames behaviours as risk factors rather than inherently risky actions.

Recommendation:

Please standardise the terminology throughout the manuscript—particularly in

- the title and abstract,

- figure and table captions, and

- the main text and Discussion.

Maintain British spelling (e.g., behaviour, behavioural) consistently across the entire manuscript.

Reviewer #5: I have reviewed the authors' responses to my revision requests, and they have satisfactorily addressed the issues.

**Do you want your identity to be public for this peer review?** For information about this choice, including consent withdrawal, please see our Privacy Policy

Reviewer #4: No

Reviewer #5: No

---

## [Author Response · Author response to Decision Letter 3]

22 Nov 2025

Editor Comments

One of reviewers requested additional work for the label of class and consistent use of terminology. Please revise.

Author response

We have revised the class labels, and used the term ‘behavioural risks’ consistently throughout the manuscript. We have also changed the title to ‘Do behavioural risks cluster among college students in Chandigarh, India? Novel insights from a latent class analysis’ as suggested by Reviewer #4.

Reviewer #4

Thank you for submitting the revised version of your manuscript. The study remains important in the context of public health and education policy, and the analytical approach is appropriate. The structure and clarity have improved compared with the previous version. However, one previously raised issue has not yet been adequately addressed, and one new point requires clarification and revision.

1. Class Labelling (Lines 217; Supplementary File S4) (Previously raised in Round 1 but not sufficiently resolved) Class 2 continues to be labelled “smoking, alcohol use and sexual risks,” which remains inconsistent with both the statistical results and conceptual interpretation.

a. Very low prevalence of sexual behaviour variables

According to the S2 File, variables related to sexual behaviour have extremely low prevalence, with the highest frequency being 21 participants (about 3%). Such a small subsample cannot provide stable class estimation; low-prevalence indicators often yield unstable parameter estimates in LCA and therefore should not define a class.

Author response

We agree that items with very low or very high overall probability have unstable estimates in latent class analysis. Indeed, some risk factors in our study, such as suicide attempts and unsafe sexual behaviours, had low prevalences.

b. Fitted prevalence patterns do not support the label

The S4 File shows that sexual-risk indicators are actually higher in Class 1 (multiple risks) than in Class 2. For example:- Condom not used at last intercourse: Class 1 = 26.6%, Class 2 = 14.6%- Four or more sexual partners in lifetime: Class 1 = 16.1%, Class 2 = 11.0%These results indicate that sexual-risk behaviours are characteristic of Class 1, not Class 2. Including “sexual risks” in the Class 2 label may therefore be misleading.

Author response

We agree that Class 1 has a higher prevalence of sexual risks than Class 2. However, it is also worth noting that the prevalence of these risk factors in Class 2 is much higher than that in Classes 3 and 4. E.g., ‘four or more sexual partners in lifetime’, Class 2: 11.0%, Classes 3 and 4: both 0.0%. Similarly, for ‘condom not used at last intercourse’: Class 2: 14.6%, Class 3: 0.4%, Class 4: 0.0%. This shows a distinct pattern of sexual risks in Class 2. (Whether these prevalence estimates are reliable is a separate issue, covered in point a. above.)

Recommendation: Please reconsider the naming of Class 2 to reflect its dominant behavioural profile more accurately—for example:- “smoking and alcohol-related risks”, or- “substance use–dominant risks.” Adding a brief statement in the Methods or Results section explaining the rationale for final class names (e.g., theoretical justification, indicator strength, or item-response probabilities) would also improve transparency.

Author response

Overall, we agree with the reviewer’s recommendation, particularly in light of point a. (low-prevalence behaviours should not define a class). Hence, we have labelled class 2 as ‘smoking and alcohol-related risks’ throughout the manuscript.

Abstract, p 2 lines 34–36:

‘Latent class analysis identified four classes: multiple risks (8.8%), smoking and alcohol-related risks (5.4%), only dietary and physical activity risks (73.7%), and victimisation and injury risks (12.1%).’

Results, p 14 line 273

“Accordingly, we labelled it as ‘smoking and alcohol-related risks’.”

We have also revised the label for Class 2 in Table 1 and S4 File.

We have now provided a brief description of the rationale and process of naming classes in the Methods.

pp 11–12, lines 209–213

‘We also assigned short, descriptive labels to each class, based on the model estimates. We labelled classes according to the behavioural risk domains which showed greatest within-class fitted prevalence. Though we had included rare risk factors (e.g., suicide attempts, sexual risks) in the models, we avoided including them in class labels, as their parameter estimates could be statistically unstable.’

2. Terminology Consistency (New observation in Round 2)

The manuscript alternates between “risky behaviours” and “behavioural risks.” Although the two expressions are related, consistent terminology is necessary for clarity and precision. Given that this study adopts an epidemiological and risk-modelling perspective, “behavioural risks” is the more technically accurate term because it frames behaviours as risk factors rather than inherently risky actions.

Recommendation: Please standardise the terminology throughout the manuscript—particularly in- the title and abstract,- figure and table captions, and- the main text and Discussion.

Author response

We agree with the reviewer’s recommendation. We have now used the term ‘behavioural risks’ consistently throughout the manuscript.

Maintain British spelling (e.g., behaviour, behavioural) consistently across the entire manuscript.

Author response

We thank the reviewer for bringing this to our attention. We have now used British spelling (e.g., ‘behavioural’ rather than ‘behavioral’) throughout the manuscript text. While referring to the CDC YRBSS, we have kept the spelling as the original American ‘Behavior’. Similarly, where American spellings come up in the References, we have kept them unchanged.

Reviewer #5

I have reviewed the authors' responses to my revision requests, and they have satisfactorily addressed the issues.

Author response

We thank the reviewer for their constructive inputs, which helped us revise this manuscript.

---

## [Decision Letter · Decision Letter 3]

16 Dec 2025

Do behavioural risks cluster among college students in Chandigarh, India? Novel insights from a latent class analysis

PONE-D-25-12897R3

Dear Dr. Lakshmi,

We’re pleased to inform you that your manuscript has been judged scientifically suitable for publication and will be formally accepted for publication once it meets all outstanding technical requirements.

Kind regards,

Yuan-Pang Wang, M.D., Ph.D.

Academic Editor

PLOS One

Additional Editor Comments (optional):

All comments have been addressed. There is improvement in consistent writing (labels) and clarification of methodology. The manuscript is technically sound to be considered for publication.

Reviewers' comments:

Reviewer's Responses to Questions

**Comments to the Author**

Reviewer #4: All comments have been addressed

2. Is the manuscript technically sound, and do the data support the conclusions?

Reviewer #4: Yes

3. Has the statistical analysis been performed appropriately and rigorously?

Reviewer #4: Yes

4. Have the authors made all data underlying the findings in their manuscript fully available?

Reviewer #4: Yes

5. Is the manuscript presented in an intelligible fashion and written in standard English?

Reviewer #4: Yes

Reviewer #4: This revised manuscript shows clear improvement and adequately addresses the major concerns raised in the previous review. The relabelling of Class 2 as “smoking and alcohol-related risks” is appropriate and better reflects the dominant behavioural profile, avoiding over-interpretation of low-prevalence sexual risk indicators. The added explanation of class-labelling principles in the Methods section improves methodological transparency and strengthens the interpretation of the LCA results.

The issue of terminology inconsistency has also been resolved. The manuscript now consistently uses the term “behavioural risks” and applies British spelling throughout, with appropriate exceptions for referenced instruments and cited literature.

Overall, the revisions enhance conceptual clarity and analytical coherence. I recommend acceptance after minor revision, subject only to final editorial checks.

**Do you want your identity to be public for this peer review?** For information about this choice, including consent withdrawal, please see our Privacy Policy

Reviewer #4: No

---

## [Editor Report · Acceptance letter]

PONE-D-25-12897R3

PLOS One

Dear Dr. Lakshmi,

I'm pleased to inform you that your manuscript has been deemed suitable for publication in PLOS One. Congratulations! Your manuscript is now being handed over to our production team.

Kind regards,

on behalf of

Dr. Yuan-Pang Wang

Academic Editor

PLOS One